# Nanowire-Based Materials as Coke-Resistant Catalyst Supports for Dry Methane Reforming

**Apolo Nambo [1,2], Veerendra Atla [1,2], Sivakumar Vasireddy [2], Vivekanand Kumar [2], Jacek B. Jasinski [1], Sreedevi Upadhyayula [3] and Mahendra Sunkara [1,***

[1]  Conn Center for Renewable Energy Research and Department of Chemical Engineering,
University of Louisville, Louisville, KY 40292, USA; apolo.nambosalgado@louisville.edu (A.N.);
veerendra.atla@louisville.edu (V.A.); Jacek.jasinski@louisville.edu (J.B.J.)
[2]  Advanced Energy Materials, LLC., 311 E Lee Street, Louisville, KY 40208, USA;
siva.vasireddy@advancedenergymat.com (S.V.); vivekanand.kumar@gmail.com (V.K.)
[3]  Department of Chemical Engineering, Indian Institute of Technology Delhi, New Delhi 110016, India;
sreedevi.upadhyayula@chemical.iitd.ac.in
*  Correspondence: mahendra@louisville.edu

**Abstract:** In this paper, nanowire-supported catalysts loaded with nickel are shown to be coke resistant compared to nanoparticle-supported catalysts. Specifically, Ni-loaded titania-based nanowire catalysts were tested with the dry methane reforming process in a laboratory-scale continuous packed-bed atmospheric reactor. The $CO_2$ conversion rate stayed above 90% for over 30 h on stream under coke-promoting conditions, such as high flow rates, low temperatures, and a high ratio of $CH_4$ to $CO_2$. The coke ($C_xH_y$, x>>y) on the spent catalyst surface for both nanowire- and nanoparticle-supported catalysts was characterized by TGA, temperature-programmed reduction (TPR), and electron microscopy (SEM/TEM/EDS), and it was revealed that the types of carbon species present and their distribution over the morphology-enhanced materials played a major role in the deactivation. The $CO_2$ conversion activity of Ni supported on titania nanoparticles was reduced from ~80% to less than 72% in 30 h due to the formation of a graphitic coke formation. On the other hand, Ni particles supported on nanowires exhibited cube-octahedral morphologies, with a high density of non- (111) surface sites responsible for the increased activity and reduced graphitic coke deposition, giving a sustained and stable catalytic activity during a long time-on-stream experiment.

**Keywords:** titania; $CO_2$; DMR; nanowire; methane; coke

## 1. Introduction

Global warming is one of the greatest problems facing humanity and it threatens society's well-being, challenges the process of economic development, and alters the natural environment [1]. Carbon dioxide is the most significant anthropogenic greenhouse gases that contributes to global warming and climate change [2]. The increase in carbon dioxide is mainly due to the burning of fossil fuels such as oil, coal, and natural gas [3]; cement fabrication is another important anthropogenic source of $CO_2$. Annually, 7 gigatons of $CO_2$ are released into the atmosphere, surpassing by far the rate of carbon capture mechanisms found in nature and leading to an accumulation of this gas over the years. One of the most promising ways to transform $CO_2$ is through the use of the dry methane reforming (DMR) reaction. During this reaction, $CO_2$ and $CH_4$ are combined to produce syngas, which is a valuable precursor for many other downstream applications [4]. Although methane is present at a lower concentration than $CO_2$, its global warming potential is 21 times greater [5]. During DMR, the aforementioned gasses are subjected to high temperatures to activate the molecules and break the C-H and C-O bonds, which are very stable [6].

$$CH_4 + CO_2 \Leftrightarrow 2CO + 2H_2 \qquad G_{1000^oK} = -24.17 \frac{KJ}{mol}. \tag{1}$$

The syngas produced is considered a building block that can be used as a reactant for other applications, such as Fischer–Tropsch fuel, methanol, and other valuable liquid fuels and chemicals [7]. This technology allows the mitigation of the most important environmental issue at present and provides a new generation of fuels [8].

The industrial application of this technology has been limited due to the lack of an effective catalyst [9]. One of the challenges associated with optimum catalytic performance is that the dry methane reforming process is highly endothermic and needs to be performed at temperatures above 700 °C, which is energy-intensive and raises operational costs [10]. However, the main challenges for dry methane reforming catalysts are stability, activity, and coke formation [9]; therefore, an efficient, economic, and stable catalyst during reaction conditions is crucial for the industrial implementation of dry methane reforming [10]. A suitable catalyst for reforming would catalyze the reaction at low temperatures, would be resistant to coke formation, and would be tolerant to different concentrations of poisons (e.g., sulfur, halogens, heavy metals, etc.) for an extended period of time [11]. Typically, nickel provides highly active sites but suffers from fast deactivation [12] from carbon deposition; such deposition is usually caused by $CH_4$ decomposition and/or CO disproportionation [13].

There have been several prior attempts to develop nickel-based catalysts with improved stability against sintering and coking. First and foremost, titania support alone did not help, but titanates had some beneficial effect [14]. Other studies have focused on supporting small particles of nickel on a variety of high-surface-area supports to increase dispersion and reduce the sinterability. These studies include supporting on silica coronas [15,16], ex-solutions [16] in pyrochlores [13], mesostructured supports [17], perovskites [18,19], porous substrates by the atomic layer deposition process (ALD), Mg-rich Mg-Al Hydrotalcite-supported Ni catalyst modified with lanthanum, and citric acid (CA) calcined at 1000 °C (La.Ni(CA)/$Mg_{1.3}AlO_x$.1000) [20]. The best time-on-stream data to date were obtained using nickel supported on a MgO support. Ruckenstein et al. [21] attribute the long-term (120 hrs) stable activity of their NiOMgO catalyst to the effect of the MgO-inhibited CO disproportionation reaction on the formation of the NiOMgO solution due to the similar crystalline structures of the two oxides. Additionally, Mo infused with nickel nanoparticles on a highly crystalline-fumed MgO support exhibited a high activity and stability.This catalyst [22] stability is attributed to the migration of the nanoparticles; to heating; and to the step edges of the support forming larger and highly stable nanoparticles, passivating the coking sites on the oxide support. Hence, it is evident that the size, structure, and nature of the support of DRM catalysts significantly affect the length of their stable activity. So far, no work has been reported on the use of nanowire-based supports for improving durability with coking resistance and reduced sinterability.

The use of nanowire has been implemented to favor certain reaction pathways and enhance the selectivity towards the desired product in these structure-sensitive reactions. Supports [23] with nanowire morphology could enhance the yield of the desired product; this feature of nanowire-based catalyst is very interesting, and it is explored within this study. The formation of $CH_x$ species on the active metal sites and their further oxidation reactions with surface carbonate species formed by the reaction of $CO_2$ with surface basic sites prevents the transformation of $CH_x$ to larger carbonaceous species. In the present work, nanowire morphologies are investigated as supports for Ni in the DMR process in terms of the type of coke formation and level of stability achieved compared to that achieved with nanoparticle-supported catalysts.

## 2. Results and Discussion

Potassium titanate nanowires, shown in Figure 1a, have dimensions ranging from 100 nm to 1 μm in diameter and 5 μm to 30 μm in length. Catalysts are made by decorating potassium titanate nanowires with nickel (with a loading of 1%) and are tested for $CO_2$ conversion in the dry methane reforming reaction. As shown in Figure 1b, the nickel-decorated nanowire sample displayed a stable conversion during the 5 h experiments

when compared with commercial potassium titanate nanoparticles. The commercial Ni-decorated potassium titanate nanoparticles had a higher initial activity, but within 3 h the activity dropped below that of the nanowire-based material.

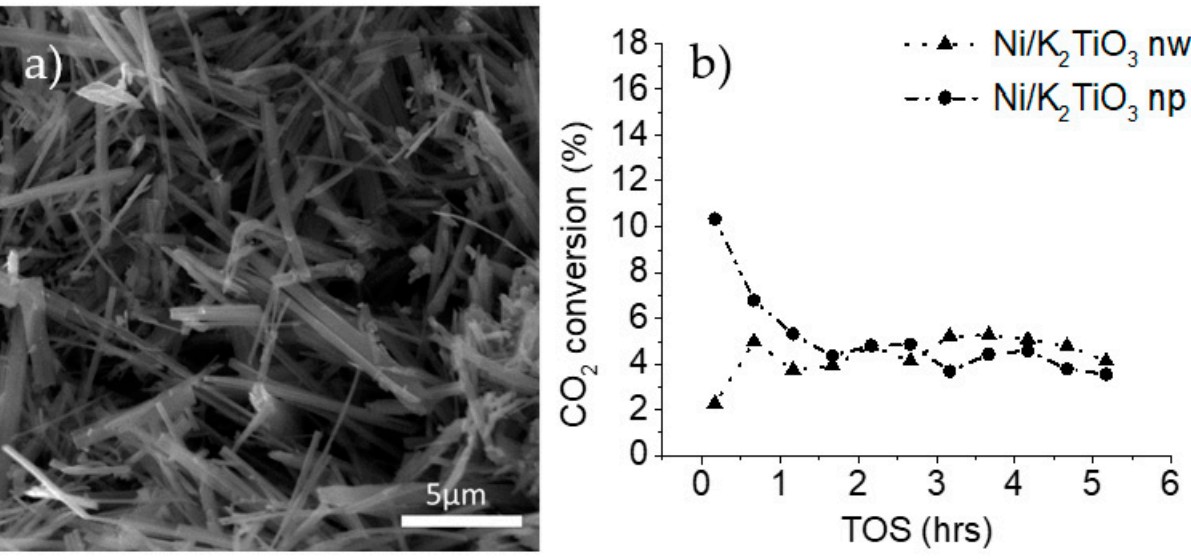

**Figure 1.** (**a**) SEM image of potassium titanate nanowires used as the support for nickel, (**b**) $CO_2$ conversion during dry methane reforming using potassium titanate as the support (750 °C, gas hourly space velocity (GHSV) of 30,000 mlgr$^{-1}$h$^{-1}$, feed ratio of methane to carbon dioxide of 2).

Titania nanowires were obtained after the acid wash of the potassium titanate nanowires. The morphology was confirmed by the SEM images below, showing that the nanowire morphology was unaltered (lengths ranging from 5 to 30 μm and diameters from 100 nm up to 1 μm). The titania nanowires were decorated with Ni in the same fashion as the potassium titanates.

When the materials were tested under the same conditions as the potassium titanates, their activity was almost 10 times greater, reaching carbon dioxide conversions higher than 70% across the 5 h test. The coke deposition and the nature of the carbon deposits can be evaluated when the materials display an appreciable deactivation. Zhang et al. reported a high coke formation using an 8 wt% nickel loading [24]. Catalysts were prepared using an 8 wt% loading of nickel using commercially obtained titania particles and titania nanowires. The catalysts were tested in the dry methane reforming reaction. The methane-to-carbon dioxide ratio was increased to 3 and the feed consisting solely of methane and carbon dioxide increased the total amount of carbon deposits. Additionally, an increase in the gas hourly space velocity (GHSV) is suggested by Jeon et al. to rapidly generate coke over dry methane reforming catalysts [25]. The GHSV was changed from 30,000 to 60,000 mlgr$^{-1}$h$^{-1}$. Under these new conditions, which were prone to causing coke formation, the nickel-decorated titania-based materials were tested and the data were collected through a time on stream (TOS) of 30 h, as shown in Figure 2c.

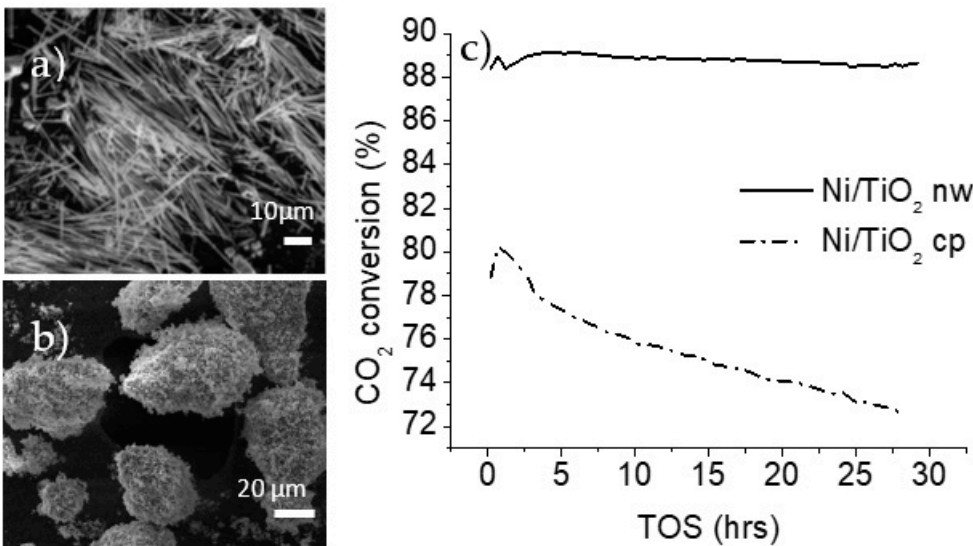

**Figure 2.** SEM images of (**a**) decorated titania nanowires (nw), and (**b**) decorated titania commercial particles (cp); (**c**) $CO_2$ conversion during dry methane reforming over titania supports (750 °C, GHSV of 60,000 mlgr$^{-1}$h$^{-1}$, feed ratio of methane to carbon dioxide of 3).

The results from the catalytic testing showed a clear difference in the behavior of nanoparticles and nanowire-supported catalysts. The carbon dioxide conversion is higher in the case of nanowire morphology, reaching an 89% conversion, which was sustained for the entire 30 h duration. The nickel-decorated titania particles had a maximum carbon dioxide conversion of 80%, and this decreased steadily throughout the testing period.

XRD patterns of both catalyst systems were taken before and after the reaction. The XRD patterns of both the as-synthesized and the spent titania particles are displayed in Figure 3a. The titania particle samples have a $TiO_2$ rutile phase in both fresh and spent catalysts. The as-synthesized sample displayed nickel oxide reflections at around a 2θ angle of 37 to 43°. In the spent catalyst, the nickel was reduced, the reflections of nickel oxide disappeared, and reflections of metallic nickel (at around a 2θ angle of 45 to 53°) appeared.

There is a noticeable reflection at around a 26° 2θ angle that corresponds to carbon graphite reflection; this reflection is broad and can accommodate reflections of other carbon formations such as carbon nanotubes. In the case of nanowire morphologies, the material showed an anatase phase that has low crystallinity, as can be seen from the XRD in Figure 3b. In the spent nanowire-based catalyst, there is a phase transformation in the support going from a low crystalline anatase to a more crystalline anatase accompanied by a rutile phase. It has been reported in the literature that a similar effect in titania nanorods, a phase transformation from anatase to rutile at 750 °C [26], can occur. The nickel shows a similar transition from nickel oxide to reduced nickel from the synthesized catalyst through the spent samples in both the nanowire-and particle-based catalyst.

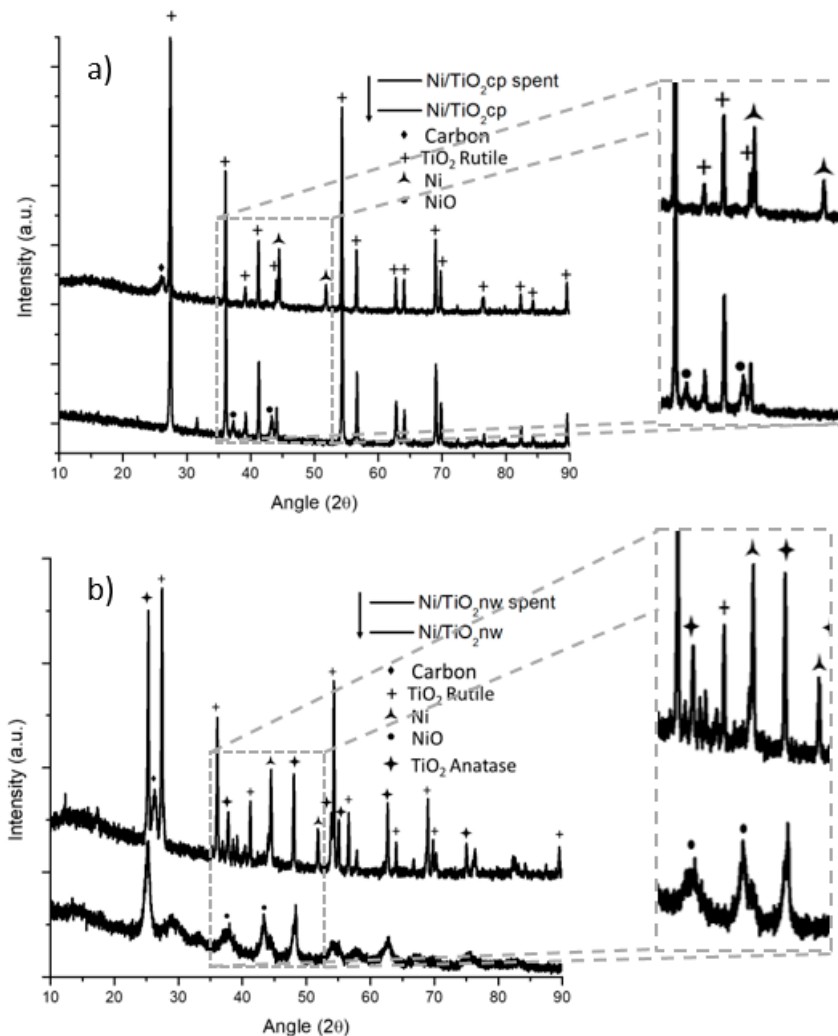

**Figure 3.** XRD of nickel-decorated titania prepared and tested in the dry methane reforming process; (**a**) commercial particle support and (**b**) nanowire support.

The carbon reflection at around a 26° 2θ angle seemed sharper in the case of the titania nanowire support, indicating a difference in coke deposits. From the results above, the thermogravimetric analysis (TGA) was performed on the spent materials to quantify and characterize the carbon deposits formed in both supports. The TGA from the nickel-decorated titania particles is reported in Figure 4a. For the analysis, a starting weight of approximately 11 mg was placed in a porcelain cup and measured using an SDT-Q600. The sample was heated up to 1000 °C using a heating rate of 10 °C per minute under 100 mL per minute of airflow. The solid line in Figure 4a is the weight of the sample as a function of the temperature. The weight of the sample started to decrease at around 508 °C and stabilized at a constant weight of 9.07 mg at around 726 °C. The 1.94 mg of weight loss represented 17.6% of the initial weight. This weight loss was coke being burned from the catalyst. The nominal temperature for the coke elimination was determined to be 637 °C based on the derivative of the weight change (gray line).

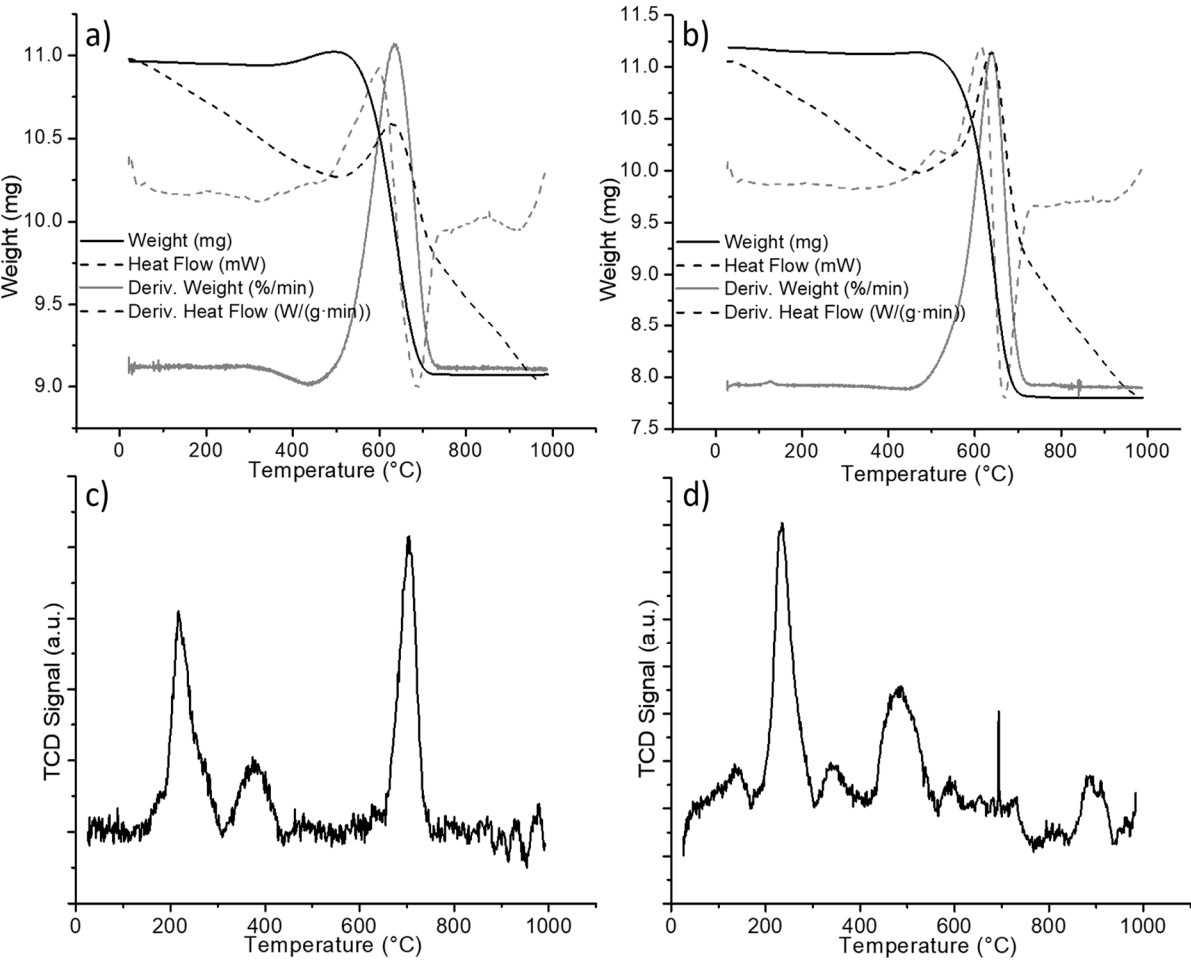

**Figure 4.** Spent nickel-decorated titania (**a**) TGA of commercial particles, (**b**) TGA of titania nanowires, (**c**) temperature-programmed reduction (TPR) of commercial particles, and (**d**) TPR of titania nanowires.

The thermogravimetric analysis for the nickel-decorated titania nanowires was performed in the same fashion. Figure 4b shows the curves obtained from the data analysis. With an initial weight of 11.13 mg, the weight loss of the sample started at 486 °C and finalized at around 721 °C, achieving a stable weight of 7.81 mg. From the derivative of the weight change curve, the nominal temperature for the coke elimination was determined to be 644 °C. The weight difference was 3.32 mg, which corresponded to 29.8% of the initial weight. When comparing the carbon elimination processes of both spent catalysts, it is evident that, for the nanowire morphology, the carbonaceous residues are eliminated at a lower temperature than in the case of particle titania supports. The thermogravimetric analyses include the heat flow curves and derivate of heat flow (dashed lines in Figure 4a,b). From the latter, it can be appreciated that in the case of nickel-decorated titania nanowires (Figure 4b), the carbon elimination presented two thermal stages; the first one was a small peak occurring around 509 °C.

The nature of carbonaceous residues can be further identified by temperature-programmed reduction (TPR). Carbonaceous residues that are easily reduced are believed to help in a decoking process that leads to CO formation; this surface carbide is denominated α-C and displays reduction temperatures of between 300 and 310 °C. β-C is reduced in the range of 580 to 600 °C and is attributed to amorphous carbon, which contributes to deactivation mechanisms [27]. Any carbonaceous residue that is reducible at higher temperatures is attributed to very stable graphitized carbon that participates in deactivation mechanisms and is classified as γ-C [28,29].

The TPR profiles are shown in Figure 4c,d, where it can be seen that the titania particle support exhibits a different distribution of carbon residues than the nanowire-based catalyst. The TPR profiles displayed a low-temperature reducible carbonaceous material ($\alpha$-C) below 300 °C. Temperatures from 300 to 600 °C are determined to correspond to $\beta$-C reduction, and above 600 °C correspond to the very stable graphitized $\gamma$-C. For the quantification of carbon types, the TPR profiles were integrated and the relative content was determined from the area ratios. The distribution of carbon-type residues according to their reducibility is reported in Table 1.

**Table 1.** Distribution of different types of carbon residues present.

| Carbon Type | Commercial Titania Support | Titania Nanowire Support |
|:---:|:---:|:---:|
| $\alpha$-C | 42.1% | 56.3% |
| $\beta$-C | 15.3% | 42.8% |
| $\gamma$-C | 42.5% | 0.9% |

When comparing the relative content of the very stable $\gamma$-C of both spent catalysts, it is clear that the elevated graphitized carbon content on the titania particles played a major role in the deactivation process. The transmission electron microscopy (TEM) images from the as-synthesized and spent catalyst showed interesting details about the morphological effects on the carbon deposits.

For the particle-based catalyst (Figure 5), the as-synthesized materials showed well-dispersed nickel oxide nanoparticles of around 10 nm in diameter. In the spent particle-based catalyst, the nickel nanoparticles displayed an increase in diameter reaching around 50 to 100 nm. It is evident that the nickel-dispersed nanoparticles are sintered after the high temperatures required for the reaction (750 °C). Moreover, a large amount of carbon in the form of aggregates of thick interconnected graphitic shells was observed in the spent samples. Although many of the shells were found empty, it was evident that they had nucleated and grown on the surface of the nickel particles.

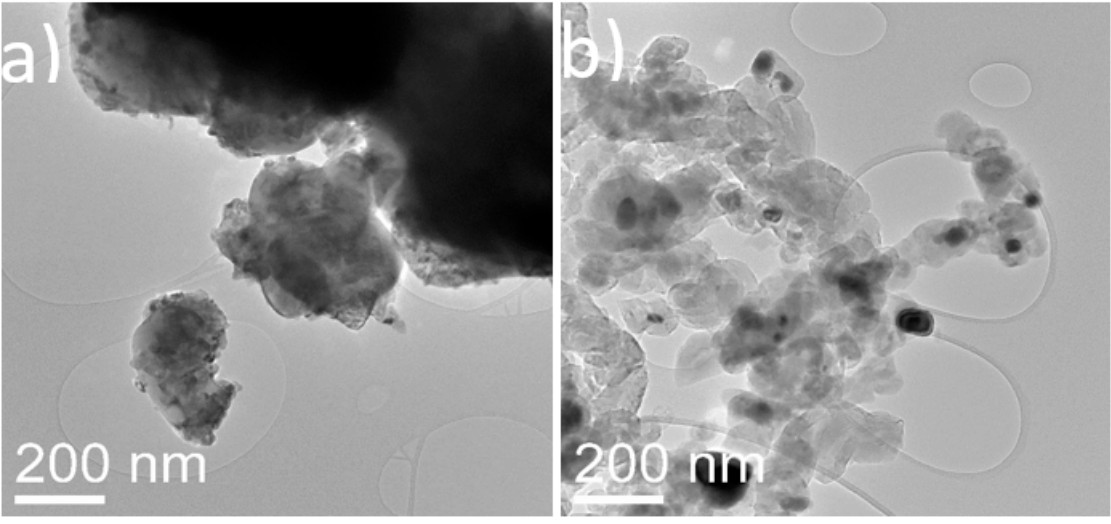

**Figure 5.** TEM images of (**a**) fresh nickel oxide supported on commercial titania powder and (**b**) spent nickel supported on commercial titania powder.

In the case of nanowires, the reduction process at reaction temperatures exhibited several changes. See Figure 6 for observations of fresh and spent nanowire-supported catalysts. First, the porous anatase nanowires transformed into rutile-phase but solid nanowires. Second, the reduced nickel formed well-defined crystals decorated on highly crystalline titania nanowires. The sizes of metallic nickel particles were distributed around

20 to 50 nm. The surfaces of nickel particles supported on nanowires did not show any presence of carbon layers. Only occasional nickel nanoparticles detached from nanowires were found with carbon shells. On the other hand, the spent catalyst made using commercial titania powder exhibited almost all nickel particles covered with graphitic layers. The growth of carbon deposition and the continuous agglomeration of metallic nickel particles supported on spherical titania nanoparticles seemed to reduce its catalytic activity.

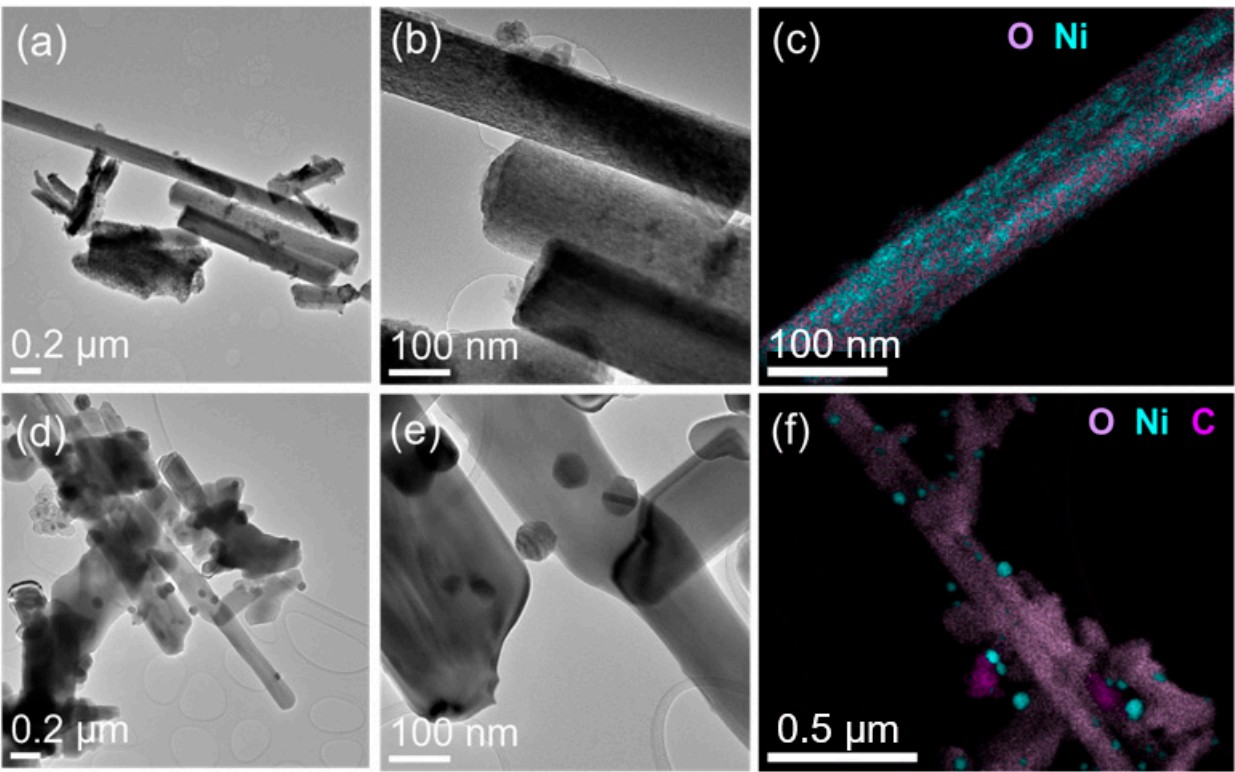

**Figure 6.** (**a**,**d**) Low- and (**b**,**e**) high-magnification TEM images and (**c**,**f**) EDS elemental maps of fresh and spent nanowire catalyst sample.

Ni supported on titania nanowire supports (8% by wt) has also been tested at stoichiometric feed conditions—i.e., 1:1 for $CH_4$ to $CO_2$. In order to reduce the phase transformation of the porous titania nanowire supports, they were first heat-treated at 700 °C for two hours prior to loading them with nickel. Such treatment partially improved the crystallinity, but the catalyst still contained an anatase phase. The catalyst sample was tested using stoichiometric feed composition and at temperatures from 750 to 800 °C. See the results in Figure 7. Results show that $CO_2$ conversion was reaching close to 100% with a slight decrease over time, stabilizing at around 99%. The lower methane conversion could be due to a favored methanation reaction which is part of the reaction network involved in this process. The XRD of the fresh and spent materials showed the full transition of the support phase from anatase to rutile. When analyzing the type of carbon deposits that are formed under these conditions, a TGA analysis revealed that the weight loss started at a lower temperature (460 °C) than when a coke-promoting ratio was used (486 °C). A total weight loss of 6.1 wt% (1.26 mg) was developed between 460 °C and 726 °C. The coke elimination is estimated to have a nominal temperature of 642 °C. When compared with the results from the feed containing a higher methane-to-carbon dioxide ratio, the nature of the coke formed behaved in the same fashion. As can be seen in Figure 7c, there is a broader peak related to the coke decomposition with a clear shoulder towards lower temperatures indicating that the results with the nature of coke formed are similar to those tested at coke-promoting conditions.

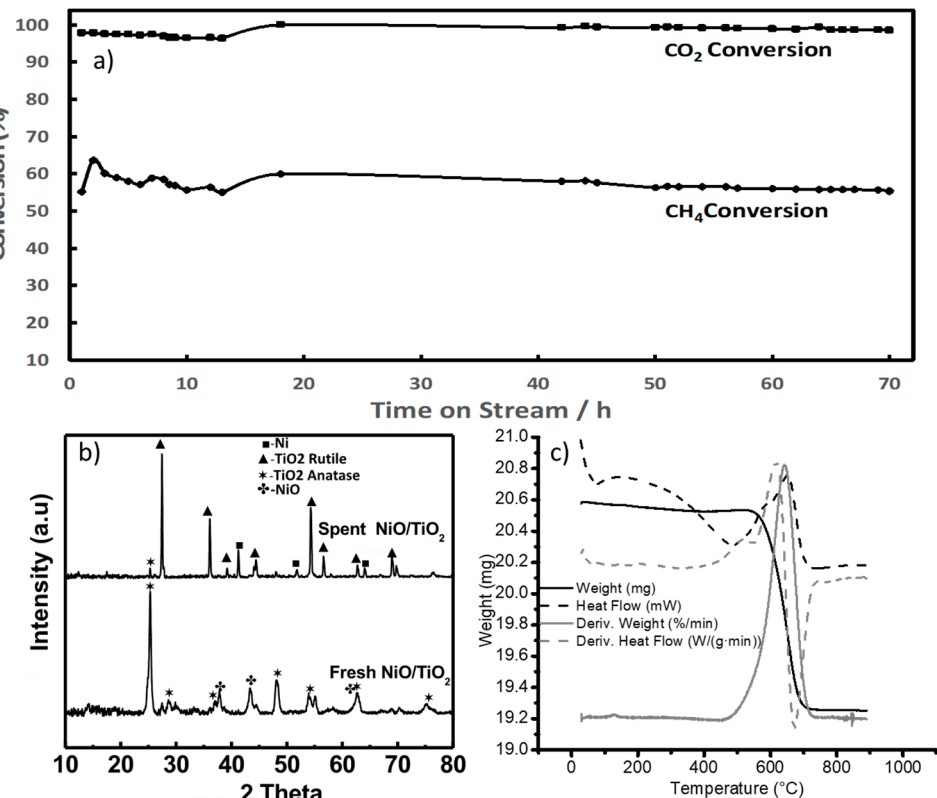

**Figure 7.** (**a**) $CO_2$ conversion during dry methane reforming over titania nw supports (750 °C–800 °C, GHSV of 60,000 mlgr$^{-1}$h$^{-1}$ (with 25% being nitrogen), equimolar feed ratio of methane to carbon dioxide. (**b**) XRD of nickel-decorated titania nw, fresh and tested in the dry methane reforming. (**c**) TGA of spent nickel-decorated titania nw.

As for the nanowire supported catalyst, the nickel particles developed a stronger interaction with the nanowire surface. High-resolution images and diffraction analyses, shown in Figure 8, indicate that the nickel nanoparticles may have an epitaxial relationship with the underlying titania nanowire surface. Additionally, some of the nickel nanoparticles are anchored at the titania surface steps (see Figure 8a). The strong interaction between nickel nanoparticles and the underlying titania nanowire prevents them from sintering. Most importantly, the particles exhibit crystalline surfaces for good reactivity. Nickel metallic particles show the presence of nickel oxide shells on their surfaces which could have formed during the reaction shutdown.

The observed differences, in terms of carbon formation as well as catalyst durability, may originate from the differences in the nickel nanoparticle surfaces and strains between the particle-and nanowire-based systems. It is well known that Ni(111) surfaces match the graphene lattice very well [30–32] and promote the growth of graphitic carbon. While the Ni(111) surfaces are in general available for the randomly oriented Ni nanoparticles in the particle-based catalysts, they may be restricted in the Ni nanoparticles which have a specific epitaxial relationship with the underlying titania nanowires. Additionally, the surface energetics of the Ni nanoparticles on such nanowires may be modified due to lattice mismatch and interfacial strains, leading to the additional suppression of the carbon shell formation. Further investigation into the phase transition of $TiO_2$ supports is required to fully understand its role in carbon suppression mechanisms.

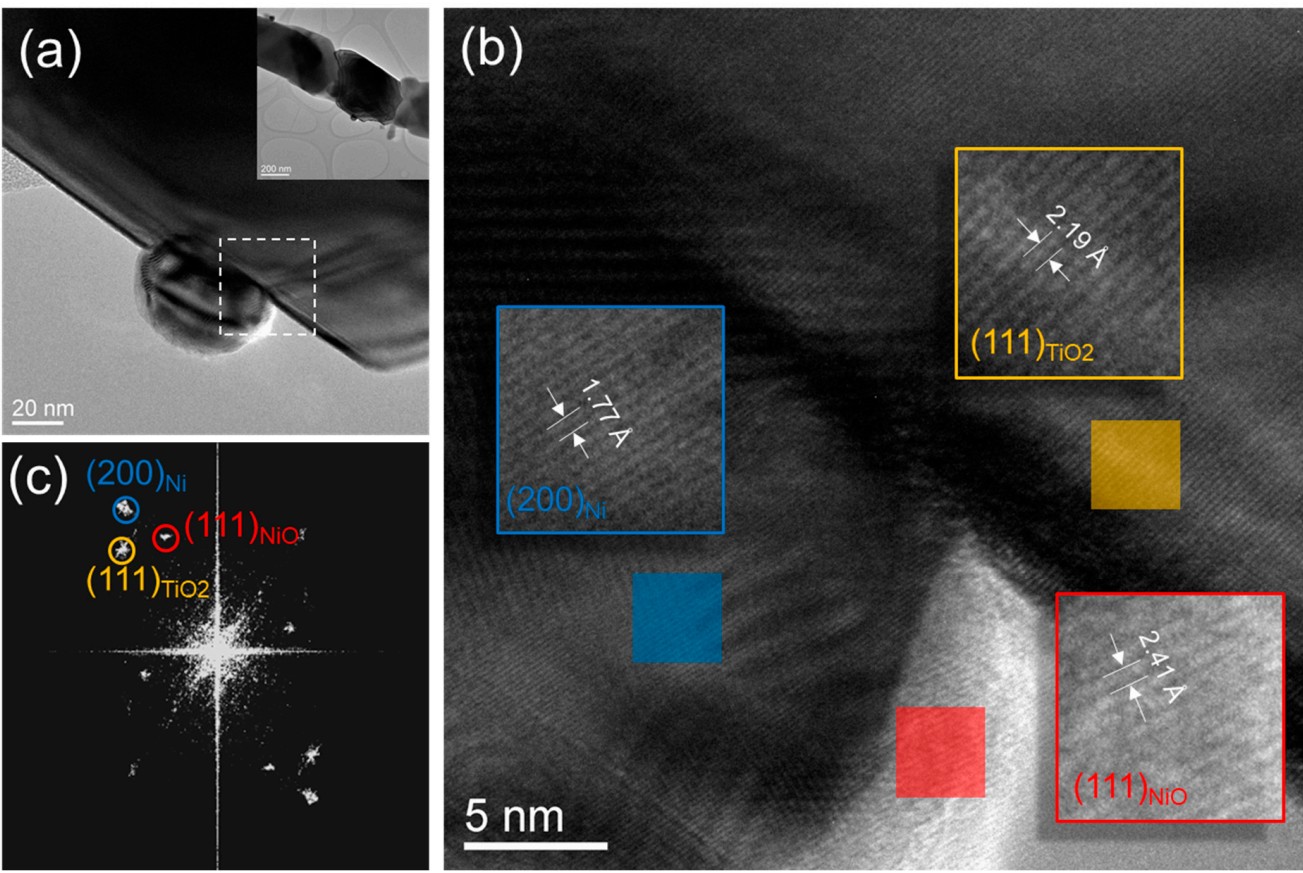

**Figure 8.** (**a**) TEM image of Ni nanoparticles on TiO$_2$ nw in the spent catalyst. A low magnification of this segmented nanowire is shown in the inset. (**b**) high resolution TEM image of the interfacial area shown with a dotted square line in (**a**). (**c**) Fast Fourier Transform (FFT) of the high resolution TEM image shown in (**b**). Reflection spots corresponding to TiO$_2$ nw, Ni np, and NiO shells are marked using yellow, blue, and red circles, respectively.

## 3. Experimental Section

Potassium titanate nanowires were synthesized using an industrial scale-up of the Solvo-Plasma$^{TM}$ technique. The powder precursors were mixed with water, and the mixture was then oxidized using a patented procedure [33]. The potassium titanate nanowire powders were further treated by acid wash, dried, and calcined to obtain titania nanowire powders. For comparison tests, the nanoparticles of titania and potassium titanate were obtained commercially from Sigma Aldrich (Darmstadt, Germany). The incipient wetness impregnation methodology was followed for the catalyst preparation. Nitrate precursors of the metallic salts were dissolved in a small amount of water. The amount of water needed was quantified according to the support used. In order to facilitate the dissolution of the salts, the solution was heated up. The homogeneous solution was added droplet-wise to the nanowire-based powder and mixed thoroughly. Once the solution addition was completed, the powder was dried overnight at 120 °C. Once dried, the powder was calcined in a box furnaced at 450 °C for 4 h. The catalysts were collected and stored for testing and characterization.

The scanning electron microscopy (SEM) imaging and energy-dispersive x-ray spectroscopy (EDS) analysis were performed in a TESCAN Vega 3 scanning electron microscope (Fuveau, France). Transmission electron microscopy (TEM) imaging was performed in a 200-kV FEI Tecnai F20 FEG-TEM/STEM device (Hillsboro, Oregon, USA). X-ray diffraction (XRD) patterns were collected using a Bruker D8 Discover diffractometer with Cu Kα radiation at 40 kV and 40 mA of accelerating voltage and current using a step size of 0.02 theta/step and a step time of 1 step/sec (Billerica, MA, USA). The thermogravimetric

analysis (TGA) was performed with an SDT Q600 instrument from TA Instruments (New Castle, DE, USA) using a ramp of 10 °C/min and an air flow rate of 100 mL/min. The temperature program reduction (TPR) was obtained in a ChemiSorb 2720 from Micromeritics (Norcross, GA, USA); the sample was degassed in He flow for 2 h, and the TPR was obtained in 10% $H_2$ in Ar at a heating rate of 10 °C/min.

The dry methane reforming was evaluated in a custom-made packed-bed reactor. A 1/2 inch diameter quartz tube was used as the reaction chamber. The quartz tube was placed in a vertical tubular furnace and the gas composition was controlled by the mass flow controllers (MFC) at the inlet, MFC's are from MKS (Andover, MA, USA) that were calibrated for He, $H_2$, $CO_2$, and $CH_4$. First, the catalyst was reduced in He with a gas flow of 20% $H_2$ at the reaction temperature of 750 °C for 2 h. After this time, the gas composition was changed to a mixture of $CO_2$ and $CH_4$ and the reaction time began; the first sampling was made at 10 min and subsequently every 30 min. Samples were taken by the sampling valve incorporated in the GC. The GC system was an Agilent (Santa Clara, CA, USA) 7820A with a thermal conductivity detector (TCD) and a flame ionization detector (FID) with a ShinCarbon ST micropacked column (100/120 mesh, 2 m, 1/16in. OD, 1.0 mm ID) provided by VWR (Radnor, PA, USA). The GC analysis was done under a flow of 10 mL/min of He as the carrier gas and heated following a set heating program (40 °C for 2 min, 10 °C/min heating up to 220 °C which was sustained for 3 min). Once the reaction test was completed, the reactor was cooled down and the catalyst was collected and stored for further analysis.

## 4. Conclusions

In this work, the dry methane reforming reaction was performed using several titania-based supports. The nickel supported on titania nanowires maintained a stable activity of >90% $CO_2$ conversion with the dry methane reforming reaction for more than 30 h at 750 °C, with a GHSV of 60,000 mlgr$^{-1}$h$^{-1}$ and a feed ratio of methane to carbon dioxide of 3. In comparison, nickel supported on commercial titania powder (spherical particles) exhibited an activity of less than 70% in the initial stages with a continuously decreasing activity over time. The characterization of spent catalysts revealed that nickel supported on commercial titania nanoparticles exhibited a higher amount of carbon deposition, with the majority of it in the graphitized $\gamma$-C. The nanowire-supported catalyst also exhibited the presence of carbon deposition, mostly in the form of $\alpha$ and $\beta$ forms, which can be etched during a reaction. Most importantly, the percentage of $\gamma$-C in the nanowire-supported catalyst was less than 1%, which is probably the reason for the observed stability. Studies also revealed that the porous anatase nanowires underwent phase transformation to become solid rutile-phase nanowires. The nickel oxide layer was transformed into well-defined nickel particles with an epitaxial relationship with underlying nanowires. Such an interaction between nickel particles and nanowire surfaces helped with the exposure of non-(111) surfaces, which could have played a role in reducing the graphitic carbon deposition and the increased activity with the dry methane reforming reaction.

**Author Contributions:** Conceptualization, S.U. and M.S.; Data curation, A.N., V.A. and J.B.J.; Formal analysis, A.N., S.V., J.B.J., S.U. and M.S.; Investigation, A.N., V.A., S.V., V.K., Jacek Jasinski and S.U.; Methodology, A.N., S.V., V.K. and S.U.; Project administration, M.S.; Supervision, M.S.; Writing—original draft, A.N.; Writing—review & editing, A.N., S.V., J.B.J., S.U. and M.S. All authors have read and agreed to the published version of the manuscript.

**Funding:** This research was funded by DOE through the SBIR program with the award number DE-SC0019939 and APC was funded by the Conn Center for Renewable Energy Research, UofL.

**Conflicts of Interest:** The authors declare no conflict of interest.

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
