# Peer review of "Nanowire-Based Materials as Coke-Resistant Catalyst Supports for Dry Methane Reforming"

_catalysts, doi:10.3390/catal11020175_

Round 1

Reviewer 1 Report

The significance of the phase transition of TiO2 nanowires from Anatase to Rutile during reaction, is mentioned, but not fully explored.

The reporting of results requires consistency throughout; for example the abstract states ‘CO2 conversion stayed above 89% over 30 hours’ whereas the conclusion this is described as ‘>90% CO2 conversion’.

Consider making the publication more concise.

Author Response

Comment 1: The significance of the phase transition of TiO2 nanowires from Anatase to Rutile during reaction, is mentioned, but not fully explored.

Response:As our original anatase TiO2 nanowires are porous, the phase transformation at the reaction temperatures significantly impacts the way nickel is arranged. In order to reduce this effect, we first tried to treat the original porous titania nanowires at reaction temperature for few hours and then loaded them with nickel. Such treatment helped with consistent performance with DMR reaction activity and durability at stoichiometric conditions. More work needs to be performed to understand the phase transformation effect on nickel particles supported and the actual reaction activity and durability. More experiments are needed to understand the impact of phase transformation on activity and durability through the resulting nickel particle size distribution. These experiments will be reported in near future.

Comment 2: The reporting of results requires consistency throughout; for example the abstract states ‘CO2 conversion stayed above 89% over 30 hours’ whereas the conclusion this is described as ‘>90% CO2 conversion’.

Response: Made revisions to the manuscript and corrected inconsistencies.

Comment 3: Consider making the publication more concise.

Response: At this stage, the manuscript only contained important experiments, data and their analysis. Making it more concise will be difficult.

Reviewer 2 Report

This paper describes the use of nickel catalyst supported on titania nanowires. The materials are characterized by several analytical techniques (before and after catalysis) giving strong experimental evidence that the support make the nickel particles more resistant to deactivation. TGA and TPR show that the carbonaceous materials deposited on nickel are more easily eliminated, probably because the interaction between nickel and titania reduces the formation of γ-C.

There are a few questions, which can be better clarified and several small flaws in the English

In the introduction, it would be convenient to add the balanced equation of the dry reforming, with the associated ΔG° (at 700 °C)

Line 39: rupture?

Line 59: coronas

Line 60: structured support

Line 74 desired product

Line 190: at a lower ….

Line 237: Ni-supporterd at….. supports

Line 237 and fig. 7: The conversion of CO2 is almost double of that and CH4. Does this means that the selectivity in H2 is low? What are the by-products?

Line 239-242: 700 °C, 750 to 800 °C

Line 312, ref 6: the title is wrong

Author Response

Responses:

Reviewer 2 suggested many editorial mistakes which we corrected in the revised manuscript.

Reviewer 2 had two main comments:

Comment 1: In the introduction, it would be convenient to add the balanced equation of the dry reforming, with the associated ΔG° (at 700 °C)

Response: We added the balanced equation and the corresponding delta G at 700C in the revised manuscript.

Comment 2: Line 237 and fig. 7: The conversion of CO2 is almost double of that and CH4. Does this means that the selectivity in H2 is low? What are the by-products?

Response: Reviewer is correct that the low conversion of methane indicates that there are additional reactions happening. There are two additional reactions that could contribute to low methane conversion: 1) methanation reaction, i.e., CO2 and H2 combine to form methane and water; 2) etching of solid carbon with hydrogen to form methane. Both these reactions contribute to low net conversion of methane compared to CO2. Additional discussion is added to the revised manuscript.